# Spin-EPR-pair separation by conveyor-mode single electron shuttling in Si/SiGe

Tom Struck[1,2], Mats Volmer [1], Lino Visser [1], Tobias Offermann [1], Ran Xue [1], Jhih-Sian Tu[3], Stefan Trellenkamp[3], Łukasz Cywiński [4], Hendrik Bluhm [1,2] & Lars R. Schreiber [1,2] ✉

Long-ranged coherent qubit coupling is a missing function block for scaling up spin qubit based quantum computing solutions. Spin-coherent conveyor-mode electron-shuttling could enable spin quantum-chips with scalable and sparse qubit-architecture. Its key feature is the operation by only few easily tuneable input terminals and compatibility with industrial gate-fabrication. Single electron shuttling in conveyor-mode in a 420 nm long quantum bus has been demonstrated previously. Here we investigate the spin coherence during conveyor-mode shuttling by separation and rejoining an Einstein-Podolsky-Rosen (EPR) spin-pair. Compared to previous work we boost the shuttle velocity by a factor of 10000. We observe a rising spin-qubit dephasing time with the longer shuttle distances due to motional narrowing and estimate the spin-shuttle infidelity due to dephasing to be 0.7% for a total shuttle distance of nominal 560 nm. Shuttling several loops up to an accumulated distance of 3.36 μm, spin-entanglement of the EPR pair is still detectable, giving good perspective for our approach of a shuttle-based scalable quantum computing architecture in silicon.

Silicon-based electron-spin qubits show single- and two-qubit gate[1–7], as well as readout[8,9] fidelities reaching the prerequisite for topological quantum error correction[5]. This pronounces the need to increase the number of spin-qubits on a chip in an architecture which does preserve the qubit's manipulation and readout performance. New qubit readout strategies[10,11] and ideas for architectures with sparse[12] and dense[13–15] qubit-grids have emerged. Sparse qubit grids have good perspective to eliminate qubit cross-talk issues of their dense counter-part[16] and to solve the signal-fanout problem[17] by employing tiles of on-chip control-electronics[12,18,19]. Sparse qubit architectures require high-fidelity coherent spin couplers that can bridge distances of several micrometers. One type of coupler involves high-impedance superconducting resonators, which necessitate a complex interface between spin and the electrical-dipole[20,21]. Other demonstrations focus on spin-qubit shuttling of one spin-qubit towards another qubit across an array of tunnel-coupled static quantum dots (QDs) named bucket-brigade shuttling[22–25]. This approach, however, is complicated by the sensitivity of adiabatic Landau-Zener transitions to potential disorder in the quantum well[26].

In this respect, spin shuttling using a moving QD-referred to as conveyor-mode shuttling-is more scalable, as it requires only four easily tunable input signals, independent of its length[26,27]. While coherent spin shuttling preserving entanglement has been demonstrated with surface acoustic waves in piezoelectric materials[28], an array of top-gates connected to four gate sets can induce a moving QD in a Si/SiGe one-dimensional electron channel (1DEC)[26]. A spin qubit shuttle device (SQS), also called QuBus, employing the conveyor-mode shuttling in Si/SiGe has been demonstrated, with a shuttle distance of 420 nm and a charge shuttling fidelity of $(99.42 \pm 0.02)\%$[27]. Subsequent improvements pushed the cumulative shuttle distance to 19 μm with a charge shuttling fidelity of $(99.7 \pm 0.3)\%$[29].

[1]JARA-FIT Institute for Quantum Information, Forschungszentrum Jülich GmbH and RWTH Aachen University, Aachen, Germany. [2]ARQUE Systems GmbH, Aachen, Germany. [3]Helmholtz Nano Facility (HNF), Forschungszentrum Jülich, Jülich, Germany. [4]Institute of Physics, Polish Academy of Sciences, Warsaw, Poland. ✉e-mail: lars.schreiber@physik.rwth-aachen.de

Here, we go one step further and characterise the spin-coherence of a SQS operated in conveyor-mode. To probe the spin-coherence, we initialize the SQS by creating a spin-entangled Einstein-Podolsky-Rosen (EPR)-pair at one end. Since the EPR-pair represents a simple example of a fully entangled two particle state[30,31], it is ideal to probe the coherence properties of our shuttling procedure. We separate the EPR-pair by conveyor-mode shuttling at a variable distance and velocity and combine them to detect the preservation of the spin-entanglement by Pauli-spin blockade (PSB).

Compared to previous work[27], we increased the shuttle velocity by four orders of magnitude to 2.8 ms$^{-1}$ while preserving the charge shuttle fidelity at $(99.72 \pm 0.01)$ % over a distance of nominal 560 nm in total. By observing coherent oscillations from singlet (S) to unpolarised triplet ($T_0$) during the shuttle process, we demonstrate the coherence of the shuttled spin-qubit up to a cumulative distances of nominal 3.36 μm. The dephasing time $T_2^*$ of the EPR-pair is initially on par with $ST_0$ dephasing in a tunnel-coupled double quantum-dot (DQD) in Si/SiGe with a natural abundance of isotopes[32]. We observe an increase of $T_2^*$ with the shuttle distance, which demonstrates the predicted enhancement of the dephasing time of the shuttled qubit by motional narrowing[26]. This motional narrowing is caused by averaging out quasistatic noise of the spin's Zeeman splitting due to its motion, leading to an increased spin-dephasing time[26,33].

## Results

### Device Layout and Method

First, we introduce the SQS device and the experimental methods. The three metallic (Ti/Pt) gate-layers of the SQS device (Fig. 1a) are isolated by conformally deposited 7.7 nm thick $Al_2O_3$ and fabricated by electron-beam lithography and metal-lift off on an undoped Si/Si$_{0.7}$Ge$_{0.3}$ quantum well with natural abundance of isotopes similar to Ref. 27. The 1DEC of the SQS is formed in the Si/SiGe quantum well by an approximately 1.2 micron long split-gate with 200 nm gate spacing (purple in Fig. 1a). Seventeen so called clavier gates are fabricated on top with 70 nm gate pitch. Eight gates are fabricated on the second gate layer labelled P1, P8, 3 × S1 and 3 × S3. Nine gates are on the third layer labelled B1, B2, B8, B9, 3 × S2 and 2 × S4. Characteristic for our SQS in conveyor mode, the shuttle gates S1, S2, S3, S4 each represent one of the four gate-sets containing two to three clavier gates. Clavier gates of one gate set are electrically connected and thus always on the same electrical potential[26,27]. Since every fourth clavier gate is on the same potential within the shuttle section, the period $\lambda$ of the electrostatic potential is 280 nm. The SQS contains two single electron transistors (SETs) at both ends which are used as electron reservoir and proximate charge sensors sensitive to the electron filling at the ends of the SQS. Due to a broken clavier gate B8 on the right side of the device, only the left side of the SQS is used.

### Pulse sequence

Figure 1b shows the simplified sequence for a shuttling experiment (details in the method section). It starts with loading four electrons from the left tunnel-coupled SET into the SQS. Then, we decouple this electron reservoir by raising B1, such that the four electrons are trapped in the first QD confined by gates B1 and B2. Next, we form a DQD under P1 and S1 with B2 controlling the inter-dot tunnel coupling. We initialise the electron system to a spin-singlet state by waiting in $(n, m) = (4, 0)$ (stage I) for approximately 1 ms, where $n$ and $m$ are the electron filling numbers of the left and right QD, respectively. Then, we adiabatically pulse to the (3,1) charge state (stages I → S) and close the DQD's tunnel barrier via B2 (S → T in Fig. 1b, c).The electron in the right QD forms a spin-singlet with the remaining three electrons. We load four electrons into our system to enhance the energy splitting between singlet and triplet states and thus increase the PSB region in gate space (Fig. 1c)[16]. The analogy to the two-spin EPR-pair is reasonable, since the simple picture holds that two of the three electrons fill one valley-orbit shell and the remaining electron is in a singlet state with the electron in the right QD[34].

Afterwards we initiate the electron shuttling process by applying sinusoidal voltage pulses on the shuttle gates S1-S4 (see details in the method section). During shuttling, the three electrons remain confined in the outermost left QD and only the separated electron is shuttled in a moving QD. After shuttling forward and backward by the same distance (Fig. 1b), we increase the tunnel coupling within the DQD again and tune the DQD into PSB (stages T → S → P in Fig. 1b, c). In this way, only the EPR pair in singlet state can tunnel into (4,0) charge state. For all three triplet states this charge transition is energetically forbidden. Finally, we close the barrier once more to freeze the charge state[35] (stage F in Fig. 1b, c) and read it out by the current $I_{SET}$. A detailed explanation of the pulse stages S, T, P and F is given in Supplementary Fig. 4.

### Coherent shuttling

In this section, we demonstrate coherent shuttling by measuring $ST_0$ oscillations as a function of shuttle velocity $v_S$, distance $d$ and two values of the global magnetic fields $B$ (Fig. 2a, b). For each measurement of the singlet probability $P_S$, 50000 shuttle cycles are evaluated. Note that the QD shuttles a distance $d$ always twice, forward and backward. We apply a simple sinusoidal signal of frequency $f$ to the gates S1, S2, S3 and S4 (see method section Charge Shuttling for the details), thus the shuttle velocity should be approximately constant and the electron shall be in motion throughout the entire shuttle process, from initialisation to readout. The total shuttle time $\tau_S$ is adjusted by varying the shuttle velocity $v_S = f\lambda$. The maximum velocity is $v_{max} = 2.8$ ms$^{-1}$ and the amplitude of the sinusoidal signals is chosen to be in the regime of large charge shuttling fidelity $\mathcal{F}_C = (99.72 \pm 0.01)$ % across a shuttle distance $d = \lambda$ (see methods section Charge Shuttling). We managed to extend this distance to $d = 1.2\lambda = 336$ nm finally limited by a drastic drop in electron return probability. The upper bound of $v_S$ does not allow to access data points

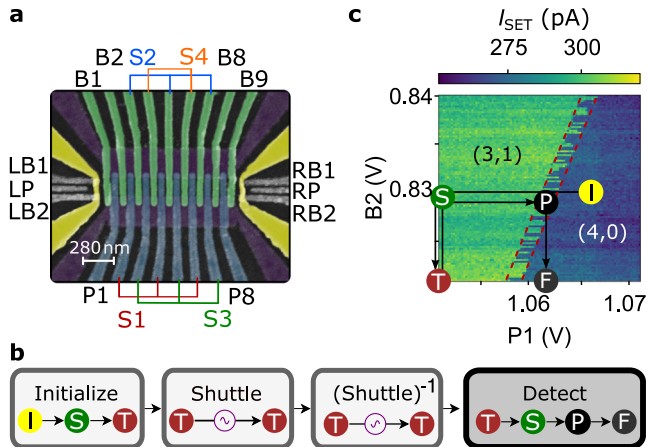

**a**

B2 S2 S4 B8
B1 B9

LB1 RB1
LP RP
LB2 RB2

280 nm

P1 P8
S1 S3

**b**

Initialize → Shuttle → (Shuttle)$^{-1}$ → Detect

**c**

$I_{SET}$ (pA)
275 300

0.84

(3,1)

0.83 S → P → I

(4,0)

T F

1.06 1.07
P1 (V)

B2 (V)

**Fig. 1 | SQS device and experimental method. a** False-coloured scanning electron micrograph of the device used in the experiment showing a top-view on the three metallic layers (1st purple, 2nd blue, 3rd green) of the SQS and their electrical connection scheme. At both ends are single-electron transistors (SETs) formed in the quantum well by gates LB1, LB2, and LP (RB1, RB2, and RP, respectively) on the second gate layer with the current path induced by the yellow gates on 3rd layer. **b** Typical voltage-pulse sequence for a shuttling experiment, separated into the EPR-pair initialisation, shuttling of one qubit forward and backward and the entanglement detection. **c** Charge stability diagram recorded by the left SET current $I_{SET}$ with labels for the absolute electron filling of the outermost left DQD of the SQS. The red dotted lines indicate boundaries of the PSB region. Labelled circles indicate voltages on B2 and P1 and correspond to the ones of **b**. Arrows indicate pulse order.

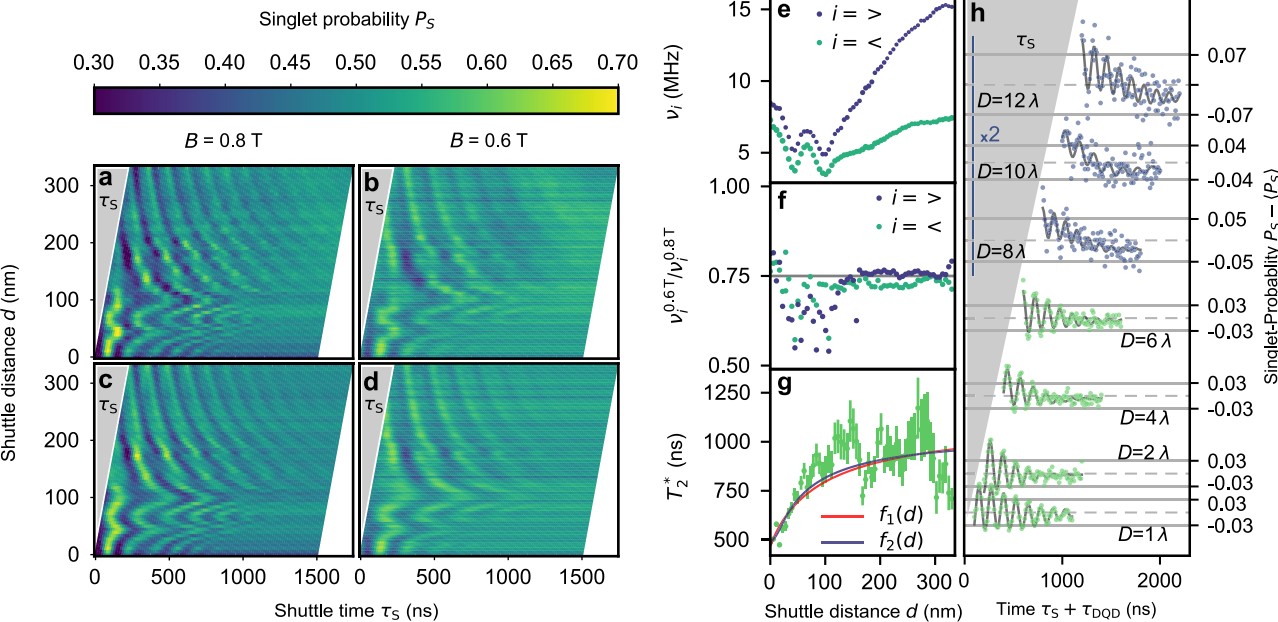

**Fig. 2 | Demonstration of coherent shuttling by EPR-pair separation and joined spin-singlet detection. a, b** ST$_0$-oscillations as a function of the shuttling time $\tau_S$ and the qubit shuttle distance $d$ at $B = 0.8$ T (panel a) and $B = 0.6$ T (panel b), respectively. **c:** Least-square fit to the data in panel **a**. **d:** Least-square fit to the data in **b**. **e** Frequencies extracted from the fit in **c**. The two frequencies are plotted with their corresponding $a_{<,>}$ encoded in the dot size. **f:** Ratio of the upper and lower frequency for each magnetic field. **g** Ensemble spin dephasing time $T_2^*$ of the EPR

pair as a function of shuttling distance with $1\sigma$-error. Red and purple lines represent least-square fits of two different fit functions. **h:** Singlet probability $P_S$ recorded after shuttling for $0.5D$ periods of cumulative distance. The time $\tau_S + \tau_{DQD}$ is the sum of the shuttle time $\tau_S$ and the time of recording the ST$_0$-oscillations in the DQD $\tau_{DQD}$. For clarity, traces are offset and amplitude is scaled for purple data points ($D = 8\lambda$, $D = 10\lambda$ and $D = 12\lambda$), for which a zoom-in version is plotted in Supplementary Fig. 2.

in the grey triangular areas (labelled with $\tau_S$) of Fig. 2a, b at small $\tau_S$ and large $d$.

We fit each line of measured ST$_0$ oscillations for both $B$ (Fig. 2c, d) with

$$P_S(\tau) = e^{-\left(\frac{\tau}{T_2^*}\right)^2} \left(a_< \cos(2\pi\nu_< \tau_S + \varphi_<) + a_> \cos(2\pi\nu_> \tau_S + \varphi_>)\right) + c, \quad (1)$$

where $P_S$ is the probability of detecting the EPR pair in a singlet state, $T_2^*$ is the ensemble dephasing time of the EPR pair, $a_{<,>}$, $\nu_{<,>}$, $\varphi_{<,>}$ and $c$ are the visibility, frequency, phase and offset of the ST$_0$-oscillations, respectively. Variations in the offset $c$ may arise from singlet initialisation and detection errors and randomly fluctuates among scanlines. We empirically find that the data can be best fitted by two oscillations, hence the two cosine terms with their respective frequencies and phases are used. We speculate that this might result from initialising a mixed valley state, as there has been two slightly different spin resonances observed in the presence of a mixed valley-state before[34,36]. Our fits (Fig. 2c, d) match the measured raw-data in Fig. 2a, b well.

First, we discuss the fitted $\nu_{<,>}$. The origin of the measured ST$_0$-oscillations is the Zeeman energy difference between the spin in the shuttled QD and the spin in the static QD, which is filled by three electrons. The difference originates from slightly different electron $g$-factors $\Delta g$ and Overhauser-energies $\Delta E_{hf}$ due to hyperfine contact interaction[37]. This is the same mechanism that leads to ST$_0$ oscillations in the case of a DQD without any conveyor-mode shuttling. These oscillations, which are effectively at $d = 0$ nm, are discussed in the method section about Singlet-Triplet oscillations. The dynamics of the nuclear spins is slow compared to a shuttle pulse sequence, but the Overhauser field might vary along the 1DEC. The electron $g$-factor depends on

valley state and QD confinement and might vary for the moving QD along the 1DEC as well[34]. Hence, the Zeeman energy difference of the entangled spins and thus the ST$_0$ oscillation $\nu_i$ frequency depends on the position $x$ of the moving QD. As this position is changing during the shuttle process, the frequency $\nu_i(d)$ becomes a function of shuttle distance $d$ and it is given by an average over the shuttling distance $d$:

$$\nu_i(d) = \frac{1}{hd} \int_0^d dx \left[\Delta g(x)\mu_B B + \Delta E_{hf}(x)\right], \quad (2)$$

where $h$ is the Planck constant, and $\mu_B$ is the Bohr magneton. We idealize by neglecting the time-dependence of $\Delta E_{hf}$ and $\Delta g$ and by assuming a deterministic thus reproducible trajectory $x(t)$ of the shuttled QD, when averaging over several shuttling cycles. Due to the integral, we expect that changes in $\nu_i(d)$ smooth out for increasing $d$. Indeed, we observe a shuttle-distance dependence of the ST$_0$ oscillation with a smoothing trend towards larger $d$ (Fig. 2e). Furthermore, we observe that the $\nu_{<,>}$ scale with the external magnetic field, which underlines the origin of our observed oscillations being spin-dynamics in agreement with Eq. (2). Calculating pairwise the ratios of $\nu_<$ and $\nu_>$ measured at $B = 0.6$ T and $B = 0.8$ T, we arrive close to the expected ratio of 0.75 (Fig. 2f). This demonstrates the linearity in magnetic field strength and indicates that the contribution of $\Delta E_{hf}(x)$ is small compared to the contribution of the electron $g$-factor difference. Furthermore, it shows the two oscillation components have distinct, but reproducible $\Delta g(x)$. For small $d$, the difference of $\nu_{<,>}$ is small increasing the fitting error, but deviations from the ratio 0.75 cannot be fully excluded here. The fitted $\varphi_{<,>}$ are plotted in the Supplementary Fig. 1.

## Spin-dephasing during shuttling

Most important is the evaluation of the ensemble spin dephasing time $T_2^*$ of the EPR-pair as a function of $d$, since it contains information on

the impact of conveyor-mode shuttling on the spin dephasing. We observe that $T_2^*$ increases with larger shuttle distance (Fig. 2g). Since qubit shuttling opens up new dephasing mechanisms[26], this result might be surprising at first sight, but is expected due to a motional narrowing enhancement of the shuttled qubit dephasing time[26]. We quantify the phenomenon by the fit $f_1(d)$ in Fig. 2g using

$$\left(\frac{1}{T_2^*}\right)^2 = \left(\frac{1}{T_{2,\mathrm{L}}^*}\right)^2 + \left(\frac{1}{T_{2,\mathrm{R}}^*}\right)^2 \frac{l_c}{d+l_c}. \qquad (3)$$

To incorporate the dependence of Gaussian decay $T_2^*$ of the EPR-pair on shuttle distance $d$, we use the quadratic addition of inverse $T_2^*$ times for the left (L) and right (R) electron spin and include a factor for motional narrowing for the shuttled qubit, where $T_{2,\mathrm{L}}^*$ is the ensemble spin dephasing time of the electron-spin that remains static in the outermost left QD, and $T_{2,\mathrm{S}}^*(d) \equiv T_{2,\mathrm{R}}^* \sqrt{\frac{d+l_c}{l_c}}$ represents the ensemble spin dephasing time of the forward and backward shuttled electron spin (total distance $2d$), averaging over a $d$ long spatial range of quasistatic-noise of its Zeeman-energy $E_z(x(t))$ having a correlation length $l_c$[26]. Note that the motional narrowing is independent of $v_S$ for noise being quasi-static on the time-scale of the shuttle, but it depends on the shuttled distance and thus the ensemble volume participating in averaging out the quasi-static noise. We distinguish between the static ensemble dephasing times $T_{2,\mathrm{L}}^*$ and $T_{2,\mathrm{R}}^*$, since we expect the confinement strength within the static QD to be less than in the moving QD. Our fit to the ensemble dephasing time of the EPR pair (Fig. 2g) results in $T_{2,\mathrm{L}}^* = (1110 \pm 90)$ ns, $T_{2,\mathrm{R}}^* = (520 \pm 20)$ ns and $l_c = (13 \pm 3)$ nm. This in total yields $T_{2,\mathrm{S}}^*(280\,\mathrm{nm}) = (2460 \pm 310)$ ns. This result implies that shuttled qubit increases its dephasing time by a factor of $\approx 4$ when shuttled twice across a distance of nominal 280 nm due to motional narrowing. Note that the data points in Fig. 2g tend to be lower than the fit for the largest $d$, which might be due to dephasing mechanisms induced by the shuttle process such as motion-induced valley excitations[26]. At very short shuttle distance $d$, a deformation of the moving QD might add to the change in spin dephasing time. Assuming a constant shuttle velocity, constant shape of the moving QD and only motional narrowing of $E_{\mathrm{hf}}(x)$, we derive the fit function $f_2(d)$ exhibiting a modified motional narrowing factor (Fig. 2g). Remarkably, we arrive at very similar fitting parameters (see Supplementary note 1).

**Long distance shuttling**
In order to increase the distance of shuttling, we always shuttle at a maximum velocity $v_{\max}$ and once the shuttled electron returns to the right QD of the DQD (stage S), we recorded the $ST_0$ oscillations by waiting between additional 0 to 1 μs prior to measure the EPR spin-state. We plot the spin-singlet probability $P_S$ of the EPR-pair as a function of the total time $\tau_S + \tau_{\mathrm{DQD}}$ of shuttling ($\tau_S$) and waiting ($\tau_{\mathrm{DQD}}$) (Fig. 2h). Due to the limited length of the shuttle zone, we increase the cumulative distance by shuttling in- and out for one period $\lambda$ multiple times. The total number of periods ($D$) shuttled forward plus backward is indicated on the left as the accumulated shuttle distance. For example for the trace labelled $D = 2$, the voltage pulses applied to S1-S4 are designed to shuttle the electron one period $\lambda = 280$ nm forward and same distance back towards the spin-detector. For $D = 1$, the electron is shuttled half a period forward, and the same distance back towards the detector. Strikingly, we still observe $ST_0$ oscillations for the trace labelled $D = 12$, for which the electron shuttles alternating six times forward and backward by $\lambda$ being nominally equivalent to an accumulated distance of 3.36 μm. The appearance of $ST_0$ oscillations show that the EPR-pair remained entangled after such long shuttling distance.

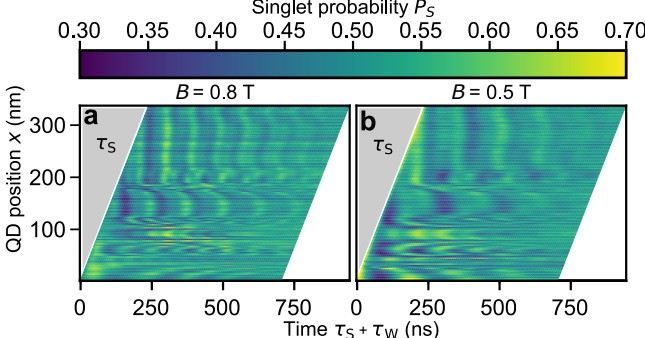

**Fig. 3 | $ST_0$-oscillations at a constant separation speed of the EPR pair as a function of shuttling distance $x$ and the total evolution time $\tau_S + \tau_W$, where $\tau_S$ is the shuttling time and $\tau_W$ is the wait time at the stationary QD position $x$.** **a** False-colour plot of the measured Singlet return probability measured at $B = 0.8$ T. **b** Same as in **a** with $B = 0.5$ T.

**Mapping local $v$ variations**
Coherent shuttling of a spin qubit and EPR separation allows us to collect information about $\Delta g(x)$ along the SQS. Instead of shuttling the spin-qubit forward and backward with a $\tau_S$-dependent $v_S$, we shuttle it by a distance $x$ along the 1DEC at maximum $v_{\max} = 2.8\ \mathrm{ms}^{-1}$, wait there for a time $\tau_W$ to let the $ST_0$ oscillations evolve and then shuttle back at maximum $v_S$ for PSB detection. We observe (Fig. 3) $ST_0$-oscillations and similar to Fig. 2a, b, their frequency $v(x, B)$ scales with the $B$-field as expected (cmp. Figure 3a and b). Compared to Fig. 2a, b, $v(x, B)$ tend to fluctuate faster as a function of $x$. This is expected, since $v_{<,>}$ results from averaging many positions $x(t)$ in the coherent shuttle experiment (Eq. (2)) in Fig. 2, while here $v$ dominantly depends on the fixed position $x$. Note that $x(t)$ and thus $d$ is not measured in any case, but deduced from the expected position of the ideal propagating wave potential $x = \lambda \frac{\Delta \varphi}{2\pi}$, where $\varphi$ is the phase of the voltages applied to gates S1-S4 relative to the initialisation potential. Hence, we neglect potential disorder and wobbling effects of the propagating wave potential, which are exemplary simulated in Ref. 26. Notably, $v(x)$ starts to become nearly constant at $x > 210$ nm. This could be an indication that the electron stops moving at this point. If we try to shuttle to $x > 330\ \mathrm{nm} > \lambda$, the electron dominantly does not return, indicating potential disorder which is sufficiently high to break the QD confinement in the propagating QD.

**Discussion**
This work shows progress on electron shuttling in conveyor-mode, building up on earlier demonstrations of charge shuttling[27]. We improved the shuttle velocity by four orders of magnitude to a regime at which coherent shuttling becomes feasible[26]. When moving into and out of the device once, we demonstrate coherent shuttling by EPR pair separation and recombination across a total distance of nominal 560 nm and at least 420 nm in case the electron spins halts at $x = 210$ nm. Furthermore, we detect entanglement when moving the electron for an accumulated shuttle distance of nominal 3.36 μm (at least 2.4 μm). Remarkably, the dephasing time of the shuttled qubit $T_{2,\mathrm{S}}^*$ is enhanced by motional narrowing, while the static electron-spin dominates the dephasing of the spin-entangled EPR-pair. Based on the fitted $T_{2,\mathrm{S}}^*(280\,\mathrm{nm}) \approx 2460$ ns ($\approx 2130$ ns for fitting with $f_2(d)$), we can estimate a phase-infidelity caused by the shuttle time $\tau_S$ at maximum shuttle velocity $v_S$ using the Gaussian decay

$$1 - \mathcal{F} = 1 - \exp\left(-\left(\frac{\tau_S}{T_{2,\mathrm{S}}^*}\right)^2\right) \approx \left(\frac{2d}{T_{2,\mathrm{S}}^* v_S}\right)^2. \qquad (4)$$

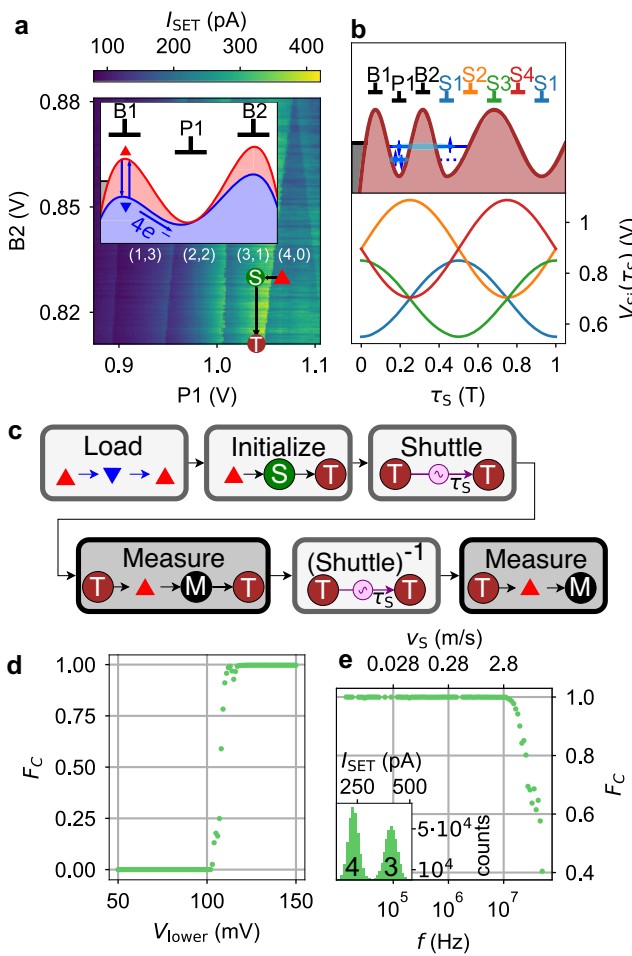

**Fig. 4 | Charge Shuttling. a** Charge scan of the DQD under gates P1 and S1 with labelled electron filling. Voltage pulse (red triangle → S → T) of the initialisation in (3,1) is marked (Stage T is at B2 = 0.7 V). Inset: Schematic of the electrostatic potential along the 1DEC under gates B1, P1 and B2 for loading. The respective electrostatic configurations are marked with red and blue triangles. **b** Shuttle Pulse. Schematic of the electrostatic potential under the labelled gates after initialisation (top). Sinusoidal voltage pulse $V_{Si}(\tau_S)$ applied to S1 – S4 to shuttle the electron one period ($\lambda = 280$ nm) forward (bottom). **c** Flowchart of the charge shuttling experiment. Labelled points correspond to panels a and b and Fig. 1b. Coloured arrows in the load and shuttle sections express that during this part of the pulse other gates than P1 and B2 are pulsed. The pulse stages M and red triangle are electrostatically the same. **d** Charge shuttling fidelity as a function of shuttle pulse amplitude $U_{lower}$ at $f = 10$ MHz. **e** Charge shuttle fidelity as a function of the shuttle pulse frequency $f$ at $U_{lower} = 150$ mV. Inset: Histogram of SET-currents measured during point M with assigned filling numbers of the QD underneath gate P1.

We estimate a shuttling-induced phase-infidelity of $1 - \mathcal{F} = (0.66 \pm 0.17)\%$ for a total shuttle distance of nominal $2d = 2\lambda = 560$ nm (at least 420 nm). Assuming a constant shuttle velocity, constant shape of the moving QD and only motional narrowing of $E_{hf}(x)$ (fit equation $f_2(d)$ see supplementary material) yields a matching infidelity of $1 - \mathcal{F} = (0.88 \pm 0.18)\%$ within the error range.

Next, we have to increase the shuttle distance by improving confinement of the moving QD. The competing electrostatic potential disorder can be reduced by replacing $Al_2O_3$ by $SiO_2$, which exhibits less interface defects, and by thinner dielectric layers[26]. We already achieved a charge shuttle fidelity of $(99.7 \pm 0.3)\%$ for total shuttle distance of 19 μm in a 10 μm long Si/SiGe QuBus[29]. The spin dephasing time can be enhanced by isotopically purified $^{28}$Si and valley excitations can be mitigated by higher valley splitting[38]. Conversely, the valley splitting can be mapped in the shuttle region[39]. Adding spin-

manipulation zones will grant more flexibility in performing coherent shuttling experiments to explore the dephasing channels and the role of the valley states. In the long run, we target at the integration of our spin shuttle device into a scalable semiconductor qubit architecture[40].

## Methods

### Charge shuttling

A prerequisite for spin-coherent shuttling is that the electron stays confined in the moving QD, which we call charge shuttling. Figure 4 depicts the pulse procedure for benchmarking the charge shuttling in the same device that we used for spin-coherent shuttling. Firstly, we load four electrons into the first QD by lowering B1 (Fig. 4a inset). Due to cross-talk, we need to compensate on P1 and B2. Thereafter, the barrier is raised again to isolate the system. Loading takes approximately 2 ms time as the voltage on B1 is 10 kHz lowpass-filtered. Subsequently, one electron is moved into the second QD (Fig. 4a, red triangle → S) and the barrier B2 is closed by pulsing it down by 120 mV (S → T). After stage T, the shuttle pulse (Fig. 4b lower part) is applied to the gate-sets S1 – S4

$$V_{Si}(\tau_S) = U_i \cdot \sin(2\pi f \tau_S + \varphi_i) + C_i. \tag{5}$$

The amplitudes $(U_1, U_3)$ applied to the gate-sets S1 and S3 on the second layer (blue in Fig. 1a) is $U_{lower} = 150$ mV, whereas the amplitudes $(U_1, U_3)$ applied to the gate-sets S2 and S4 on the 3rd metal layer is slightly higher ($U_{upper} = 1.28 \cdot U_{lower} = 192$ mV) to compensate for the difference of capacitive coupling of these layers to the quantum well. This compensation extends to the DC-part of the shuttle gate voltages. The maximum amplitude is limited to 192 mV due to the attenuation installed in the cryostat together with the maximum peak-to-peak output amplitude of the arbitrary waveform generator of 5 V. The offsets $C_1 = C_3 = 0.7$ V are chosen to form a smooth DQD, whilst $C_2 = C_4 = 0.896$ V are chosen to form a smooth DC potential. The phases are chosen to build a travelling wave potential across the one-dimensional electron channel ($\varphi_1 = -\pi/2, \varphi_2 = 0, \varphi_3 = \pi/2, \varphi_4 = \pi$). This travelling wave potential is illustrated in Fig. 4b at the top part. The barrier B2 is pinched off to limit the cross-talk-influence from the shuttle pulse to the static electrons. The electron is moved adiabatically by one period of the travelling wave potential (280 nm) to the right. After one period, the absolute gate voltages are exactly identical to the prior state, when the charge scan in Fig. 4a has been recorded. Hence, we can check whether the electron is shuttled away by going back to the electrostatic configuration corresponding to the red triangle and measuring the SET current. By time reversing the voltage pulses on S1 – S4, we shuttle the electron back and perform a measurement in a similar manner (detailed description of time reversed pulses in Supplementary Fig. 3). Then, we calculate a histogram as shown in the inset of Fig. 4e, fit two Gaussian distributions and take the fits crossing point to define the range of $I_{SET}$ assigned to three and four electron detection events. Only if the first measurement yields three (i.e. electron is shuttled away from detector) and the second measurement four electrons (i.e. electron is shuttled back to detector), a shuttling event is counted as successful. The same approach for counting successful charge shuttle events has been used in Ref. 27.

In Fig. 4d, we plot the charge shuttling fidelity $\mathcal{F}_C$ as a function of the lower layer amplitude $U_{lower}$, the upper layer amplitude is $U_{upper} = 1.28 \cdot U_{lower}$ to compensate for the larger distance to the 1DEC. We find a steep rise of $\mathcal{F}_C$ at $U_{lower} > 110$ mV. The histogram of $I_{SET}$ for all $U_{lower} > 125$ mV (inset of Fig. 4e) shows well separated Gaussians assigned to either four or three electron filling of the QD underneath gate P1. Due to nonlinear effects on the SET, the peak for four electrons is narrower than the peak for three electrons. Figure 4e shows charge shuttling fidelities as a function of shuttle frequency $f$ as defined in Eq. (5) which corresponds to a shuttle velocity $v_S = f\lambda$. From the green

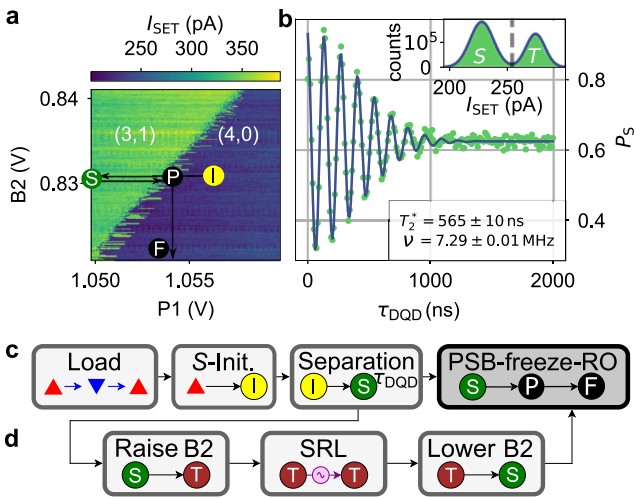

**Fig. 5 | ST$_0$-oscillations and pulse for coherent shuttling. a** Charge stability-diagram of the DQD in Fig. 4a. **b** ST$_0$-oscillation as a function waiting for time $\tau_{DQD}$ at stage S. The singlet probability $P_S$ is measured by applying the pulse from panel c spending various time $\tau_{DQD}$ at stage S. The solid line is fit to the data. inset: Readout histogram with threshold (dashed line) determined by the fitted crossing point of the two Gaussians. **c** Schematic representation of the experiment pulse. Numbers correspond to stages in **a** and Fig. 4a. **d** Schematic of the additional stages complementing the pulse scheme in **c** for doing the spin-shuttling measurement. They slot in panel c in between the separation and the PSB-freeze-RO pulse segments. The SRL section represents any applied shuttle pulse.

points we read off high fidelities up to 10 MHz (2.8 m/s), presumably limited by electrostatic disorder in the 1DEC. By averaging $\mathcal{F}_C(U_{lower} > 125\,\text{mV})$ in Fig. 4d, we calculate the mean charge shuttling fidelity for shuttling a nominal total distance of $2\lambda = 560$ nm ($\lambda$ forwards and backwards) to be $\mathcal{F}_C = (99.72 \pm 0.01)$ %. This value is slightly better than the charge shuttling fidelity of 99.42% obtained in Ref. 27. Moreover, we found charge shuttling across $2\lambda$ and back at $f = 2$ MHz. We tracked the charge by measuring the charge state after every shuttle-pulse, which moves the electron by one period $\lambda$ (cmp. shuttle tomography method in Ref. 29) and calculated a transfer fidelity of 98.7% at the same voltage amplitudes.

## Singlet-triplet oscillations

To demonstrate that the single-electron spin-qubit coherently shuttles, we use the preservation of the entanglement with the static electron spin, which we detect by the coherent oscillations between spin-singlet S and unpolarised spin-triplet T$_0$ state of this EPR pair

$$H = \begin{pmatrix} -J(\varepsilon) & \frac{\Delta g \mu_B B + \Delta E_{hf}}{2} \\ \frac{\Delta g \mu_B B + \Delta E_{hf}}{2} & 0 \end{pmatrix} \quad (6)$$

in the $(|S\rangle, |T_0\rangle)$-basis. Here, $J(\varepsilon)$ represents the exchange interaction as a function of the detuning $\varepsilon (= V_{P1})$ between the left and right QD. $\Delta g$ is the g-factor difference between the two QDs[26,34]. $\Delta E_{hf}$ is the Overhauser-energy-difference between the two dots. After loading four electrons as shown in Fig. 4a, we initialise the system to S(4,0) by waiting at stage I (Fig. 5a) for 2 ms. Next, we step $V_{P1}$ by 20 mV which reduces $J(\varepsilon)$ and turns on $\Delta g \mu_B B$ by letting one electron adiabatically tunnel into the right QD. As the two electrons are laterally separated, they are subject to different electron g-factors resulting in different Zeeman-energies as a result of the global B-field of 0.8 T. At stage S, we wait for $\tau_{DQD}$ time and pulse to the PSB in stage P where spin information is converted to charge information. The conversion takes approximately 500 ns after which a raise of the inter-dot barrier freezes the charge state for readout (stage F). Iterating over this pulse

scheme, we record the singlet return probability $P_S$ (Fig. 5b), which is fitted by

$$P_S(dt) = a \cdot e^{-\left(\frac{\tau_{DQD}}{T_2^*}\right)^2} \cos(2\pi\nu dt + \varphi) + c. \quad (7)$$

We yield for the spin dephasing time of the entangled spin-state $T_2^* = (565 \pm 10)$ ns and the frequency $\nu = (7.29 \pm 0.01)$ MHz. Figure 5c summarises the pulse in a schematic way. For coherent shuttling experiments, instead of waiting at the separation stage the sequence presented in Fig. 5d is inserted between the separation and PSB-freeze-RO pulse segments shown in Fig. 5c.

## Experimental setup

All experiments are conducted in a dilution refrigerator with a base temperature of 40 mK. All DC lines to the device are filtered by pi-filters ($f_c = 5$ MHz) at room temperature and by 2nd order RC filters with $f_c = 10$ kHz at base temperature. The clavier gates, B2, P1, P8, and B8 are connected to resistive bias-tees with a cutoff frequency of 5 Hz. Signals are applied to the AC and DC input terminal of the bias-tee, in order to allow inclusion of millisecond long pulse segments. A serial resistor is added to the low-frequency terminal, the value of which is tuned by flattening the sensor signal response. The SETs are DC-biased by 100 µV and readout by a transimpedance amplifier and an analog to digital converter.

## Data availability

The data generated in this study have been deposited in the Zenodo database (https://doi.org/10.5281/zenodo.8413694).

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

## Acknowledgements

We acknowledge the support of the Dresden High Magnetic Field Laboratory (HLD) at the Helmholtz-Zentrum Dresden - Rossendorf (HZDR), member of the European Magnetic Field Laboratory (EMFL). This work has been funded by the German Research Foundation (DFG) under Germany's Excellence Strategy - Cluster of Excellence Matter and Light for Quantum Computing" (ML4Q) EXC 2004/1 - 390534769, by the Federal Ministry of Education and Research under Contract No. FKZ: 13N14778, and by the National Science Centre (NCN), Poland under QuantERA programme, Grant No. 2017/25/Z/ST3/03044. Project Si-QuBus received funding from the QuantERA ERA-NET CoFund in Quantum Technologies implemented within the European Union's Horizon 2020 Programme. The device fabrication has been done at HNF - Helmholtz Nano Facility, Research Center Juelich GmbH[41].

## Author contributions

T.S., M.V. and L.V. conducted the experiments. T.S., M.V., T.O., L.V. and L.R.S. analysed the data. J.T. and R.X. fabricated the device. S.T. wrote the e-beam layers. Ł.C. derived motional narrowing effect of nuclear spins. L.R.S. designed and supervised the experiment. L.R.S and H.B. guided all authors. T.S., M.V., L.V., T.O. and L.R.S. wrote the manuscript, which was commented by all other authors.

## Funding

## Competing interests

Conveyor-mode shuttling is covered by a patent family (EP 4031486, US 2022/0293846 A1, CN114424346 A) by inventors L.R.S, H.B., Künne, Seidler. The patent application, co-owned by RWTH Aachen University and the Forschungszentrum Jülich, is currently pending. Qubit initialization and readout in the shuttle device is covered by patents families (US11687473B2, EP4031489, CN114402441) and (US 2022/0327072, EP4031487, CN114424344) by inventors L.R.S., H.B., Künne, co-owned by RWTH Aachen University and the Forschungszentrum Jülich. US11687473B2 is granted, all other pending. L.R.S. and H.B. are founders and shareholders of ARQUE Systems GmbH. The other authors declare no competing interest.
