## [Peer Review File · Nature Communications]

Spin-EPR-pair separation by conveyor-mode single electron shuttling in Si/SiGeREVIEWER COMMENTS

Reviewer #1 (Remarks to the Author):

Demonstration of electron charge and spin shuttling is gaining traction in spin qubit research community to scale-up semiconductor quantum processing architecture, while reducing cross talk between qubit and allowing distant qubit gate operation and error correction.

The authors describe in this manuscript a method of shuttling a single electron across a chain of 8 quantum dots using the method known as conveyor belt. A singlet state is initialised and using the conveyor belt scheme, one of the electrons is shuttled back and forth across the quantum dot chain for a fixed distance and the singlet-triplet oscillations is used to confirm the shuttling of the electron and used as a metric for the fidelity of the shuttling process.

The experiment and methods are sound. However, there are key issues in the analysis and conclusions that requires clarification. Below are major concerns that would need to be resolved for the manuscript to be considered publishable in the journal:

1. It is not entirely clear how the fidelity calculated by the authors in Section III is a metric for shuttling fidelity. Based on the expression used in Eq. (4), the fidelity is calculated based on the shuttling time (τ_S), and the characteristic time of the singlet state ($T_{(2,S)^*}$), which is a measurement of the lifetime of the singlet-triplet oscillations and is unrelated to the shuttling process itself. One would assume that the decay of the singlet-triplet oscillations depends largely on the wait time at the control point and does not depend on the shuttling process that takes the electron to the control / wait point. In short, is Eq. (4) meant to measure the fidelity of the shuttling or singlet-triplet oscillations?

2. In Fig. 2g, can the authors justify the fit for the data? It is not immediately clear that the fit is suitable for the data beyond a shuttle distance of approximately 80 nm. In addition, can the dip in the T_2^* numbers at $d = 180\text{nm}$ and beyond 300 nm be discounted as simply fluctuations in the measurements? Could it be some other secondary effect contributing to the drop in T_2^* ? Do the authors have measurements beyond the maximum shuttle distance plotted here?

3. In Fig. 2h, the data and fitting for $D = 8, 10, 12 \lambda$ are poor. Please clarify. In relation to this, do the authors have similar measurements as shown in Figs. 2a, b for the data points in Fig. 2h corresponding to $D > 4$? The argument that the electron can still be coherently shuttled up to large accumulated shuttle distances is not well supported with only Fig. 2h.

In addition, below are some minor concerns:

1. The gate material used is Ti/Pt which is not common for this type of quantum dot devices. Is there a reason for this?

2. In Fig 1 caption, "...Labelled circles indicate voltages on *B1* and P1 and correspond to the ones of panel b. Arrows indicate pulse order...". Is B1 meant to be B2?

3. In Fig 2(h), it is advisable to label the oscillations as singlet probability on the y axis.

4. Typo in page 4 of the main text, "...and the amplitude of the sinusoidal signals is chosen to be in the regime *of of* large charge shuttling fidelity..."

5. Typos in page 7 of the main text,

i. "...Then, we calculate a histogram as shown in the inset of Fig. 4*(e)*, fit two..."

ii. "...125mV (inset of Fig. 4*(d)*) shows..."

iii. "...This value *is is* slightly better than the charge shuttling..."

iv. In Fig 5 caption: "...They slot in panel c *inbetween* the separation and the PSB-freeze-RO pulse segments..."

6. In the supplementary materials, some typos are spotted:

Page 1: ...is the *Overhaused* field acting on the shuttled electron when the expectation value of its position along the channel is...

Page 1: ...The *measured* singlet return probability signal is then given by...

Page 2: ...an envelope function that is approximately constant on *lengthscale* of...

Page 3: ...(left QD is *not not* in the shuttle potential and thus much less confined),...

7. What sets the upper bound for both the amplitude of sinusoidal signals and the maximum velocity?

8. From the discussion in the manuscript, it implies that the main difficulty lies in shuttling the electron beyond 330 nm is potential disorder, and an accumulated shuttling distance of 3.36 μm is only achieved by shuttling back and forth within a particular length. Would this not be a key limiting factor when it comes to future devices, even if the length is increased? Have the authors considered possible mitigating solutions to this limitation?

9. Out of curiosity, did the authors try to shuttle the electron forward, and backward multiple times (and at different distance backward and forward) and inspect for the singlet probability?

Regards,
Kok Wai Chan
Mengke Feng
UNSW

Reviewer #2 (Remarks to the Author):

Reviewer #3 (Remarks to the Author):

The authors experimented electron-shuttling with conveyor-mode and report for the first time of its spin state. A spin singlet spin-pair was used for evaluation of spin phase acquired during shuttling one of its electrons. S-To oscillations were analyzed to compare with theoretical expectation of spin phase accumulation.

The measurement of spin state during conveyor-mode shuttling is novel and thorough analysis on its continuous movement across its channel has physics implications on the electric and magnetic magnitude variation within the channel. The results agree well with their detailed theoretical paper of Ref. 26 and has provided realistic parameters of correlation lengths, necessary to maintain coherence in the recent trend of scale-up application. The presentation of their evaluation on micron-scale shuttling length and m/s order speed are in fact a modest evaluation, and still has room to investigate the anticipated valley states after possible improvement in complete devices envisioned in near future. For its thorough and enlightening content, this paper deserves publication in Nature Communications without major corrections.

I'd like to address one point for completeness. Data on the measured phase offsets ($\phi_{<}$ and $\phi_{>}$) are lacking. Additional description on how the experimental data shapes of Fig. 2a or 3 can appear, by also pointing to the phase offset parameters, is needed to understand the result. For example, the twice discrepancy in v of Fig. 2e around $d=300\text{nm}$ is not directly recognizable from Fig. 2a or 3, and rather the two sine components look only offset without changing much its

frequency. This kind of question from misunderstanding can be avoided.

The following points are optional but welcomed to clarify for non-expert readers:

- The authors explained that the shuttled qubit increases its dephasing time by a factor of ≈ 4 when shuttled twice across a distance of nominal 280nm due to motional narrowing. Explain if it is happening only by increasing the shuttling distance or if shuttling velocity also plays a major role in motional narrowing.
- When the shuttling frequency increases more than 10 MHz, charge shuttling fidelity decreases. Explain why it decreases.
- Explain in detail about the fabrication. Compared to your previous work, the isolating layer thickness has been changed. Explain the reason. Also, explain why you chose a ~ 1.2 -micron long split gate and seventeen clavier gates for fabricating the SQS device.
- The authors have tried to shuttle the electron to $x > 330\text{nm} > \lambda$ but it has not returned due to potential disorder which is sufficiently high to break the QD confinement in the propagating QD. According to your previous simulation (ref 26), you can shuttle 10 μm with a shuttling velocity of $\geq 10\text{m/s}$. Explain what obstacles are in the way of achieving it in the experiment.
- EPR spin entanglement and motional narrowing are very important topics in this paper. Add a few lines about them for introduction in the introduction section, which may be helpful for the readers.
- In the coherent shuttling section, the authors indicated that they empirically find that the data can be best fitted by two oscillations, hence the two cosine terms with their respective frequencies and phases are used. They also speculate that this might result from initializing a mixed valley state. Provide references which match your observations. Also, explain if there are possibilities other than the mixed valley state.
- The authors indicated that by time reversing the voltage pulses on S1-S4, they shuttle the electron back and perform a measurement in a similar manner. Explain briefly with the help of a diagram. It may be helpful for the readers who will try to do the experiment in future.
- The authors indicated that the data points in (Fig. 2g) tend to be lower than the fit for the largest d , which might be due to dephasing mechanisms induced by the shuttle process such as motion-induced valley excitations. Explain whether it might be due to valley excitations of atomistic steps or a smooth interface. Provide references other than your papers. Explain what ways are available to reduce motion-induced valley excitation in the future.
- Provide the abbreviation for τ_W . It represents a waiting time to let the ST0 oscillations evolve and then shuttle back at maximum v_S for PSB detection. However, this can be challenging to grasp at first. Therefore, please include the abbreviation as well.
- As some typographical and grammatical errors are identified, proofread the manuscript.
- The left side of the SQS is only used due to a broken clavier gate B8 on the right side of the device. Show the figure of the broken gate in supplementary for reference and explain why it was broken.

Reviewer #4 (Remarks to the Author):

Dear editor,

we would like to thank you and the reviewers for the time taken to assess our manuscript as well as for the rapid and knowledgeable review. Here our point-by-point reply to the comments of the 4 reviewers. All changes are marked in red in the markup-version of our manuscript at the end of this letter.

Reviewer #1 (Remarks to the Author):

Demonstration of electron charge and spin shuttling is gaining traction in spin qubit research community to scale-up semiconductor quantum processing architecture, while reducing cross talk between qubit and allowing distant qubit gate operation and error correction.

The authors describe in this manuscript a method of shuttling a single electron across a chain of 8 quantum dots using the method known as conveyor belt. A singlet state is initialised and using the conveyor belt scheme, one of the electrons is shuttled back and forth across the quantum dot chain for a fixed distance and the singlet-triplet oscillations is used to confirm the shuttling of the electron and used as a metric for the fidelity of the shuttling process.

The experiment and methods are sound. However, there are key issues in the analysis and conclusions that requires clarification. Below are major concerns that would need to be resolved for the manuscript to be considered publishable in the journal

Response to the remarks to the author:

We thank the reviewer for pointing out the significance of shuttling for the scalability of spin qubit architectures and the mitigation of cross talk issues, which are indeed a timely issue.

We thank the reviewer for the positive evaluation of our experiment and methodology and we address all the concerns below:

1) Reviewer comment: It is not entirely clear how the fidelity calculated by the authors in Section III is a metric for shuttling fidelity. Based on the expression used in Eq. (4), the fidelity is calculated based on the shuttling time (τ_S), and the characteristic time of the singlet state ($T_{(2,S)}^*$), which is a measurement of the lifetime of the singlet-triplet oscillations and is unrelated to the shuttling process itself. One would assume that the decay of the singlet-triplet oscillations depends largely on the wait time at the control point and does not depend on the shuttling process that takes the electron to the control / wait point. In short, is Eq. (4) meant to measure the fidelity of the shuttling or singlet-triplet oscillations?

We thank the reviewer for the astute comment. The fidelity here is based on the ensemble spin dephasing time of the shuttled electron (i.e. $T_{2,S}^*$ defined on page 4 right after Eq. 3) not the dephasing time of the singlet triplet state (i.e. T_2^*) itself. We extract the former from the fit to the data in Fig. 2g. Thus, it includes the effect of motional narrowing. The fidelity depends on the ratio between the ensemble spin dephasing time during shuttling over the total shuttling time and the

assumption that the ensemble spin dephasing follows a Gaussian decay. Equation 4 is then about the fidelity of the shuttled electron only, not about the dephasing of the singlet triplet oscillation.

2) Reviewer comment: In Fig. 2g, can the authors justify the fit for the data? It is not immediately clear that the fit is suitable for the data beyond a shuttle distance of approximately 80 nm. In addition, can the dip in the T_2^* numbers at $d = 180$ nm and beyond 300 nm be discounted as simply fluctuations in the measurements? Could it be some other secondary effect contributing to the drop in T_2^* ? Do the authors have measurements beyond the maximum shuttle distance plotted here?

Response: We thank the author for the insightful comment. The two fits represent the only models we can sufficiently motivate without too many additional assumptions. The fits are intended to match the overall trend of T_2^* , not all its settled details. For the long distance regime, the fluctuations in T_2^* might be related to a short-range electron tunnelling due to electrostatic disorder superimposed with the propagating shuttle potential and partial motional-induced valley excitation (see page 4, paragraph 2). Modelling such disorder is difficult due to many additional parameters required. We want to avoid over fitting the data, but it seems interesting to look at such fluctuations in future work with more shuttle data at hand.

We were unable to measure T_2^* times beyond the shuttle distance shown here, since the electron charge could not be shuttled further at the high shuttle velocity, presumably due to larger potential disorder. Please also refer to the comment on page 7, paragraph 2. We added there:

“..., presumably limited by electrostatic disorder in the 1DEC”

3) Reviewer comment: In Fig. 2h, the data and fitting for $D = 8, 10, 12$ λ are poor. Please clarify. In relation to this, do the authors have similar measurements as shown in Figs. 2a, b for the data points in Fig. 2h corresponding to $D > 4$? The argument that the electron can still be coherently shuttled up to large accumulated shuttle distances is not well supported with only Fig. 2h.

Response: We thank the reviewer for the valuable feedback. For $D=8, 10$ and 12 , the visible oscillation amplitude is low and hard to see by eyes. However, as we state in the manuscript, the fact that we still observe ST oscillations with the correct frequency indicates that the EPR pair is still partially entangled despite the long spin shuttling. In order to display the residual ST oscillations, we plotted a zoom-in version of the curves for $D=8, 10, 12$ with the corresponding least-square fit in the Supplementary Fig. 2 and added to the caption of Fig. 2:

“, for which a zoom-in version is plotted in the Supplementary Fig. 2”

Please notice that the data in Fig. 2h and Fig. 2a is recorded in a different way and plotting line-cuts similar to Fig. 2a does not make much sense. In Fig. 2a and b the time variation is due to changing the time during shuttling τ_S (i.e. at constant distance the shuttle velocity is changed). In Fig. 2h the electron is shuttled at maximum velocity (for τ_S) and finally enter the DQD. In the DQD ST-oscillations are recorded, since these oscillations are a sensitive method to identify a partially

entangled state. Therefore, we use different labels for the x-axes in Fig. 2a, and Fig. 2h. We adapted slightly the caption of Fig. 2, to make this point clear.

4) Reviewer comment: The gate material used is Ti/Pt which is not common for this type of quantum dot devices. Is there a reason for this?

Response: We use Pt as the gate material since it forms smaller grain sizes, which ease the fabrication of narrower gates. The Ti is a common adhesion layer that sticks well to silicon. While other materials also work, these are readily available in our cleanroom.

5) Reviewer comment: In Fig 1 caption, "...Labelled circles indicate voltages on *B1* and P1 and correspond to the ones of panel b. Arrows indicate pulse order...". Is B1 meant to be B2?

Response: We thank the reviewer for the thorough reading of the text and finding this error. The mistake has been corrected in the manuscript. It now reads in the caption of Fig. 1:

"The red dotted lines indicate boundaries of the PSB region. Labelled circles indicate voltages on B2 and P1 and correspond to the ones of panel b"

6) Reviewer comment: In Fig 2(h), it is advisable to label the oscillations as singlet probability on the y axis.

Response: We thank the reviewer for the useful advice

We corrected the y-axis label in Fig 2 h.

7) Reviewer comment: Typo in page 4 of the main text, "...and the amplitude of the sinusoidal signals is chosen to be in the regime *of of* large charge shuttling fidelity..."

Response: We greatly appreciate the thorough reading of the text by the reviewer and we are thankful for the reviewer finding all the typos in the main text as well as in the supplementary material. These mistakes are corrected in the manuscript and the supplementary material.

8) Reviewer comment: Typos in page 7 of the main text,

i. "...Then, we calculate a histogram as shown in the inset of Fig. 4*(e)*, fit two..."

ii. "...125mV (inset of Fig. 4*(d)*) shows..."

iii. "...This value *is is* slightly better than the charge shuttling..."

iv. In Fig 5 caption: "...They slot in panel c *inbetween* the separation and the PSB-freeze-RO pulse segments..."

Response: We greatly appreciate the thorough reading of the manuscript by the reviewer and finding the errors. The mistakes have been corrected in the manuscript.

These sections read now:

- i) Then, we calculate a histogram as shown in the inset of Fig. 4e
- ii) 125 mV (inset of Fig. 4e) shows
- iii) This value is slightly better than the charge shuttling
- iv) They slot in panel c **in between** the separation and the PSB-freeze-RO pulse segments

9) Reviewer comment: In the supplementary materials, some typos are spotted:

Page 1: ...is the ***Overhauled*** field acting on the shuttled electron when the expectation value of its position along the channel is...

Page 1:...The ***measured*** singlet return probability signal is then given by...

Page 2: ...an envelope function that is approximately constant on ***lengthscale*** of...

Page 3: ...(left QD is ***not not*** in the shuttle potential and thus much less confined),...

Response: We again greatly appreciate the thorough reading of the manuscript by the reviewer and finding the errors. The mistakes have been corrected in the manuscript and the relevant sections now read:

- i) is the **Overhauser** field acting on the shuttled electron when the expectation value of its position along the channel is
- ii) The **measured** singlet return probability signal is then given by
- iii) an envelope function that is approximately constant on **length scale** of
- iv) (left QD is not in the shuttle potential and thus much less confined)

10) Reviewer comment: What sets the upper bound for both the amplitude of sinusoidal signals and the maximum velocity?

Response: The maximum amplitude for the sinusoidal signals is set by the attenuation installed on the RF lines in the cryostat and as such is relatively easy to increase in future experiments. We expect the amplitude to be overall limited by heating effects (see reference 26). The maximum velocity in this manuscript was likely limited by electrostatic disorder in the channel.

We have added a comment about the maximum amplitude to the manuscript (page 6, right column, middle):

The maximum amplitude is limited to 192 mV due to the attenuation installed in the cryostat together with the maximum peak-to-peak output amplitude of the arbitrary waveform generator of 5 V.

Moreover, we added a comment regarding the electrostatic disorder (page 7, left column, second paragraph):

, presumably limited by electrostatic disorder in the 1DEC

11) Reviewer comment: From the discussion in the manuscript, it implies that the main difficulty lies in shuttling the electron beyond 330 nm is potential disorder, and an accumulated shuttling distance of $3.36 \mu\text{m}$ is only achieved by shuttling back and forth within a particular length. Would this not be a key limiting factor when it comes to future devices, even if the length is increased? Have the authors considered possible mitigating solutions to this limitation?

Response: To us, there are two possible reasons for the shuttling failure. The first reason is a local defect in the potential due to disorder. We should be able to overcome any issue of this kind by simply increasing the shuttling amplitude to a point where it exceeds the disorder potential. Note for this that we have recently shown electron shuttling across a distances of $10 \mu\text{m}$ (<https://arxiv.org/pdf/2306.16375>), with more than 100 gates or 34 quantum dots in a row, where we have observed the possibility of overcoming potential disorder with increased amplitude. The second reason could be a broken clavier gate, which would lead to a shuttling failure for obvious reasons.

12) Reviewer comment: Out of curiosity, did the authors try to shuttle the electron forward, and backward multiple times (and at different distance backward and forward) and inspect for the singlet probability?

Response: The Fig. 2h, the cumulative shuttling distance is achieved by shuttling forward and backwards multiple times and the results are discussed in the main text. The resulting oscillations are measurements of the singlet probability. As for different distances than the maximum available distance, this we have not inspected.

Reviewer #2 (Remarks to the Author):

We thank the reviewer for co-reviewing the manuscript and supporting early career researchers.

Reviewer #3 (Remarks to the Author):

The authors experimented electron-shuttling with conveyor-mode and report for the first time of its spin state. A spin singlet spin-pair was used for evaluation of spin phase acquired during shuttling one of its electrons. S-To oscillations were analysed to compare with theoretical expectation of spin phase accumulation.

The measurement of spin state during conveyor-mode shuttling is novel and thorough analysis on its continuous movement across its channel has physics implications on the electric and magnetic magnitude variation within the channel. The results agree well with their detailed theoretical paper of Ref. 26 and has provided realistic parameters of correlation lengths, necessary to maintain coherence in the recent trend of scale-up application. The presentation of their evaluation on micron-scale shuttling length and m/s order speed are in fact a modest evaluation, and still has room to investigate the anticipated valley states after possible improvement in complete devices envisioned in near future. For its thorough and enlightening content, this paper deserves publication in Nature Communications without major corrections.

We thank the reviewer for pointing out the significance and novelty of our work and for the clear recommendation for publication in Nature Communications without major corrections.

1) Reviewer comment: Data on the measured phase offsets ($\phi_{<}$ and $\phi_{>}$) are lacking. Additional description on how the experimental data shapes of Fig. 2a or 3 can appear, by also pointing to the phase offset parameters, is needed to understand the result. For example, the twice discrepancy in v of Fig. 2e around $d=300\text{nm}$ is not directly recognizable from Fig. 2a or 3, and rather the two sine components look only offset without changing much its frequency. This kind of question from misunderstanding can be avoided.

Response: We thank the reviewer for the insightful comment and have added the plots (Supplementary Figure 1 (Supplement, page 7, top)) for the phases in the supplement.

We added to the manuscript (page 4, end of paragraph 1):

The fitted $\varphi_{<,>}$ are plotted in the Supplementary Fig. 1

2) Reviewer comment: The authors explained that the shuttled qubit increases its dephasing time by a factor of ≈ 4 when shuttled twice across a distance of nominal 280nm due to motional narrowing. Explain if it is happening only by increasing the shuttling distance or if shuttling velocity also plays a major role in motional narrowing.

Response: As indicated by equation 3 in the main text, motional narrowing only depends on the distance shuttled. This is because the narrowing is an effect of shuttling over an ensemble of spins, which increases only with the volume that is shuttled over.

We have added a relevant note to the paper (page 4, paragraph 2):

Note that the motional narrowing is independent of v_s for noise being quasi-static on the time-scale of the shuttle, but it depends on the shuttled distance and thus the ensemble volume participating in averaging out the quasi-static noise.

3) Reviewer comment: When the shuttling frequency increases more than 10 MHz, charge shuttling fidelity decreases. Explain why it decreases.

Response: We greatly thank the reviewer for the question. In all likelihood, the decrease in shuttling fidelity after the 10 MHz point is due to potential disorder. We have added a relevant note to the paper (page 7, paragraph 2):

, presumably limited by electrostatic disorder in the channel

4) Reviewer comment: Explain in detail about the fabrication. Compared to your previous work, the isolating layer thickness has been changed. Explain the reason. Also, explain why you chose a ~ 1.2 -micron long split gate and seventeen clavier gates for fabricating the SQS device.

Response: In general, we find it advantageous to lower the oxide thickness (see Ref. 26) as much as possible while it remains isolating to reduce the distance of the gates on top to the quantum well and potentially reduce the amount of charge defects. There is no special reason for the length of the split gate. We added to the text (page 5, paragraph 4, see also answer to comment 5).

The competing electrostatic potential disorder can be reduced by replacing Al₂O₃ by SiO₂, which exhibits less interface defects, and by thinner dielectric layers [26].

5) Reviewer comment: The authors have tried to shuttle the electron to $x > 330\text{nm} > \lambda$ but it has not returned due to potential disorder which is sufficiently high to break the QD confinement in the propagating QD. According to your previous simulation (ref 26), you can shuttle $10 \mu\text{m}$ with a shuttling velocity of $\geq 10\text{m/s}$. Explain what obstacles are in the way of achieving it in the experiment.

Response: To us, there are two possible reasons for the shuttling failure. The first could be a simple broken gate, which would lead to a shuttling failure for obvious reasons. The second reason is a local defect in the potential due to disorder. We should be able to overcome any issue of this kind by simply increasing the shuttling amplitude to a point where it can exceed the disorder potential. Compared to the simulations in Ref. 26, the dielectrics are not fully optimised in the current device. In the simulation in Ref. 26, we assumed 5 nm SiO₂ and in the current device we use 7.7 nm of Al₂O₃ having an order of magnitude more defects, thus leading to more potential disorder. Note also that we have recently shown electron shuttling over distances of $10 \mu\text{m}$ in Ref. 29, with more than 100 gates or 34 quantum dots in a row. There, we exceeded the potential disorder with increased amplitude on the shuttle gates. We added to the text (page 5, paragraph 4)

The competing electrostatic potential disorder can be reduced by replacing Al₂O₃ by SiO₂, which exhibits less interface defects, and by thinner dielectric layers [26].

6) Reviewer comment: EPR spin entanglement and motional narrowing are very important topics in this paper. Add a few lines about them for introduction in the introduction section, which may be helpful for the readers.

Response: We thank the reviewer for the insightful commentary. We have added a short introduction about motional narrowing and EPR with new references in the introduction (page 1, paragraph 4):

This motional narrowing is caused by averaging out quasistatic noise of the spin's Zeeman splitting due to its motion, leading to an increased spin-dephasing time [26, 32].

for the EPR pair (page 1, paragraph 4):

. Since the EPR-pair represents a simple example of a fully entangled two particle state [30, 31], it is ideal to probe the coherence properties of our shuttling procedure. We

7) Reviewer comment: In the coherent shuttling section, the authors indicated that they empirically find that the data can be best fitted by two oscillations, hence the two cosine terms with their respective frequencies and phases are used. They also speculate that this might result from initializing a mixed valley state. Provide references which match your observations. Also, explain if there are possibilities other than the mixed valley state.

Response: We thank the reviewer for the useful comment. We have added a citation, which has observed occupation of two valley states before together with the sentence (page 3, right column, bottom):

...that this might result from initialising a mixed valley state, as there has been two slightly different spin resonances observed in the presence of a mixed valley-state before [34,36].

We are currently working on exploring this further and are working on different explanations but we know of none that are convincing, which is why we do not comment on it in the paper.

8) Reviewer comment: The authors indicated that by time reversing the voltage pulses on S1-S4, they shuttle the electron back and perform a measurement in a similar manner. Explain briefly with the help of a diagram. It may be helpful for the readers who will try to do the experiment in future.

Response: We thank the reviewer for the comment and have added such a diagram for the time reversal to the supplements (Page 9, top). We added to the manuscript a reference (page 6, right column, bottom):

(detailed description of time reversed pulses in Supplementary Fig. 3)

Moreover, we added a diagram explaining the pulse stages S, T, P and F in order to support the understanding of the separation, transport and readout stages in the supplements (page 10, top). We added to the manuscript a reference (page 2, right column, bottom):

A detailed explanation of the pulse stages S, T, P and F is given in Supplementary Fig. 4.

9) Reviewer comment: The authors indicated that the data points in (Fig. 2g) tend to be lower than the fit for the largest d , which might be due to dephasing mechanisms induced by the shuttle process such as motion-induced valley excitations. Explain whether it might be due to valley excitations of atomistic steps or a smooth interface. Provide references other than your papers. Explain what ways are available to reduce motion-induced valley excitation in the future.

Response: We thank the reviewer for the excellent comment. In the data we have collected here, we can not differentiate between excitations due to atomistic steps and ones due to a smooth interface. The valley excitations can in future be reduced by increasing the valley splitting

We have added a sentence for this in the conclusion with a citation on future prospects to increase the valley splitting (page 5, paragraph 4)

and valley excitations can be mitigated by higher valley splitting [38]

10) Reviewer comment: Provide the abbreviation for τ_W . It represents a waiting time to let the STO oscillations evolve and then shuttle back at maximum v_S for PSB detection. However, this can be challenging to grasp at first. Therefore, please include the abbreviation as well.

Response: We thank the reviewer for the useful and valuable feedback and have added an explanation of τ_W to the caption of Fig. 3:

as a function of shuttling distance x and the total evolution time $\tau_S + \tau_W$, where τ_S is the shuttling time and τ_W is the wait time at the stationary QD position x

11) Reviewer comment: As some typographical and grammatical errors are identified, proofread the manuscript.

Response: We thank the reviewer for the commend and have corrected several mistakes (see also the comments 7,8,9 of Reviewer 1.)

12 Reviewer comment: The left side of the SQS is only used due to a broken clavier gate B8 on the right side of the device. Show the figure of the broken gate in supplementary for reference and explain why it was broken.

Response: We thank the reviewer for the comment, but as we do not have an image of the broken gate in question, we cannot provide a figure for it. We can detect the broken gate by the fact that sweeping a voltage on the gate shows no capacitive cross-coupling response on the right sensing dot, which is in contrast to the other gates around B8. As for the reason why it was broken, we cannot provide further insights.

Reviewer #4 (Remarks to the Author):

We thank the reviewer for co-reviewing the manuscript and supporting early career researchers.

REVIEWERS' COMMENTS

Reviewer #1 (Remarks to the Author):

We do not have any further comments and happy to accept the revised manuscript for publication.

Regards,
Kok Wai Chan
Mengke Feng
UNSW

Reviewer #2 (Remarks to the Author):

Reviewer #3 (Remarks to the Author):

We thank the authors for a thorough response to our review.
The authors have sufficiently answered and revised the manuscript and is ready to be published in Nature Communications.

Reviewer #4 (Remarks to the Author):
